# VAE BASED MULTI-FIELDS NEURAL DATA ASSIMILATION FOR SEA ICE MODEL

## ABSTRACT

This study presents a neural data assimilation system based on a variational autoencoder (VAE) for improving sea ice forecasts in high-resolution numerical models. We propose a multi-field assimilation approach that simultaneously processes several physical fields, leveraging a modern VAE architecture enhanced with pixel-wise self-attention mechanisms to capture complex spatial and cross-field correlations. Our method is validated using real-world satellite observations (Sentinel-3 SRAL and AMSR2) and operational data from the NEMO ocean model with integrated sea ice component (SI3). Results demonstrate that the framework effectively assimilates sparse and noisy observations, reducing errors in sea ice concentration estimates and improving forecast accuracy. Crucially, we demonstrate the compatibility of the neural assimilation solution with the NEMO restart mechanism, enabling seamless integration into operational forecasting pipelines. This work bridges the gap between machine learning-based assimilation and practical ocean modeling, offering a scalable, non-Gaussian alternative to traditional methods like 3D-VAR.

## 1 INTRODUCTION

Significant trend in Earth Sciences is the growing interest in the Arctic region, driven by the rapid decline in sea ice cover (Babb et al., 2023) and the profound environmental, economic, and geopolitical changes this entails (Kortsch et al., 2015; Shu et al., 2023; Dvoynikov et al., 2021). Numerical models in the Arctic and Antarctic regions are notoriously harder to run and calibrate (Pan et al., 2023; Allende et al., 2024). Sea ice forecasting requires considering the entangled relationships between wind, ocean physics, and plastic deformations inside the ice itself to be relevant. These forecasts hold practical value for ice-impeded navigation in addition to scientific value.

Naturally, Earth Sciences move towards increasing the resolution of climate models (Moreno-Chamarro et al., 2025; Olason et al., 2021; Selivanova et al., 2024) and expanding the volume of accumulated observational data (Copernicus Climate Change Service (C3S); National Aeronautics and Space Administration) in order to improve the prediction of the extreme events and circulations. This trend is driven by advances in computational power, improved satellite and sensor technologies.

Numerical models are a cornerstone in ocean simulation, incorporating modern knowledge about the complex physical processes that govern ocean dynamics. Because of the chaos, inherent to all such models, small uncertainty of the initial conditions grows to unpredictable model behavior in a few days horizons of simulation. Data assimilation is a necessary tool to condition these numerical models on observations and improve their quality. Classical data assimilation algorithms rely on the assumptions of linear model dynamics and Gaussian noise. However, as the resolution of the model increases, these assumptions become less valid, which requires a departure from these constraints to better capture the complexities of high-resolution systems (Carrassi et al., 2018).

The data assimilation task can be formulated as the process of iterative updating of the physical fields of the model $x_b$ (background) based on the observed data $y$. The target is to get closer to an unknown true state of the fields $x_a$ (analysis) (Bannister, 2008a;b). The data assimilation process must account for the spatial and temporal relationship of the physical fields.

There are several approaches to data assimilation. Non-neural approaches will be called in our text classical data assimilation methods. In most cases, it comes down to filtering techniques such as

Best Linear Unbiased Estimator (BLUE) and Kalman Filter (KF) (Kalman, 1960; Jazwinski, 1970; Evensen, 1994; 2003) or to the optimization of a cost function as 3 and 4-Dimensional Variational (3D-VAR, 4D-VAR) (Bannister, 2008a;b; Courtier et al., 1998; Rabier et al., 2000) data assimilation. Neural networks have shown a remarkable ability to learn complex and non-linear relationships between physical fields in oceanography (Zhao et al., 2024). There are several studies that combine traditional data assimilation algorithms and modern neural networks (Blanke et al., 2024; Arcucci et al., 2021; Penny et al., 2022; Cai et al., 2024; Tian, 2024; Hatfield et al., 2021; Mack et al., 2020; Peyron et al., 2021; Farchi et al., 2021; Barthelemy et al., 2022; Melinc & Zaplotnik, 2024).

Most existing data assimilation models focus mainly on simplified systems and benchmark examples, such as the Lorenz-63 or Lorenz-96 systems (Blanke et al., 2024; Arcucci et al., 2021; Penny et al., 2022; Tian, 2024; Peyron et al., 2021; Farchi et al., 2021). These toy models serve as foundational testbeds for evaluating the performance of assimilation techniques due to their ability to capture essential features of chaotic dynamics while remaining computationally tractable. However, their simplicity often limits the direct applicability of these methods to more complex, real-world systems characterized by high dimensionality and intricate spatial correlations.

Observation data can be sparse and contain errors and inaccuracies. The true values of the physical field are unknown, as are the true values of the model and the observation error. We should use some estimation for calculation and validation. This is especially noticeable when calculating the matrix $B$. Classical data assimilation algorithms operate under Gaussian assumptions. For example, the ice concentration field has a significantly non-Gaussian error distribution, while the Ensemble Kalman Filter EnKF was used to assimilate this field (Lisæter et al., 2003).

## 1.1 3D-VAR and 4D-VAR

This is family of iterative data assimilation algorithms that are based on cost function optimization. Cost function for 3D-VAR is

$$J(x) = (x - x_b)^T B^{-1}(x - x_b) + (y - Hx)^T R^{-1}(y - Hx) \tag{1}$$

where $B$ is a background error covariance matrix, $R$ is an observation error covariance matrix, and $H$ is the forward operator. For 4D-VAR the model operator $M$ is added:

$$J(x_0) = (x_0 - x_b)^T B^{-1}(x_0 - x_b) + \sum_{i=1}^{N} (y - H(x_i))^T R^{-1}(y - H(x_i)) \tag{2}$$

where $x_0 = x(t_0)$ are the physical fields at the beginning of the assimilation window and $x_i = x(t_i) = (\prod_{k=1}^{i} M_k)x(t_0)$. It can be shown that Kalman Filter minimizes the same function (Brasseur, 2006).

## 1.2 Artificial Neural Networks for Data Assimilation

An alternative to classical data assimilation schemes is the integration of artificial neural networks (ANNs) to replace or enhance components of traditional systems. There are two main strategies for incorporating ANNs into data assimilation. The first involves using a neural network to distill classical methods (Arcucci et al., 2021; Farchi et al., 2021; Zavala-Romero et al., 2025), such as 3D-Var, 4D-Var, and ETKF, capturing their key features in a more computationally efficient form. The second approach integrates ANNs directly into the assimilation process by replacing specific components of traditional algorithms (Penny et al., 2022; Tian, 2024; Hatfield et al., 2021; Mack et al., 2020; Peyron et al., 2021; Barthelemy et al., 2022; Melinc & Zaplotnik, 2024), potentially improving adaptability and providing more accurate error estimates. The second approach is of greater interest because it enables a more flexible and adaptive data assimilation process. By replacing key components of traditional algorithms with ANNs, we can take advantage of their ability to learn complex non-linear relationships within the data. This has the potential to improve accuracy, reduce computational costs, and enhance robustness in dynamically changing ice conditions.

Classical data assimilation algorithms face two major challenges. The first is the use of the covariance matrix, which can be extremely large and may fail to capture nonlinear interactions between variables. The second is the dynamical operator, which is either computationally expensive or based on overly simplistic approximations.

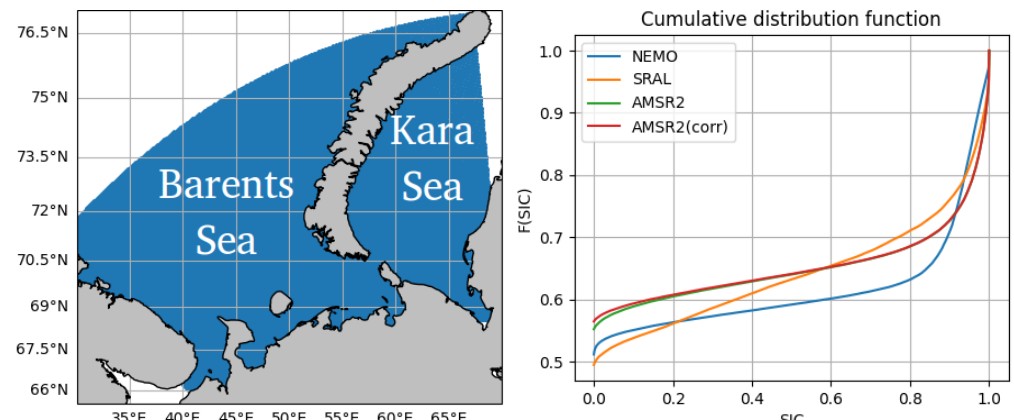

Figure 1: Area of interest for this work. Blue is the simulation area of NEMO model.

Figure 2: Cumulative distribution functions (CDFs) of sea ice concentration from different data sources.

The variational autoencoder (VAE) is widely used as a replacement or complement to the covariance matrix, enabling dimensionality reduction while preserving complex variable interactions. VAEs impose constraints on data distribution, making them particularly suitable for physical data fields with nonlinear dependencies. In (Mack et al., 2020), a VAE is applied within the 3D-Var algorithm to reduce the dimensionality of physical fields while capturing intervariable relationships, tested on pollution data from Elephant and Castle, London. In (Peyron et al., 2021), the ensemble Kalman filter is applied in the latent space of an autoencoder and tested on the Lorenz-96 model. In (Melinc & Zaplotnik, 2024), an autoencoder completely replaces the covariance matrix in the 3D-Var algorithm. The approach is tested on temperature data at the 850-hPa pressure level from the ERA5 reanalysis. The latter study is conceptually similar to our work but considers only a single physical field and does not assess the possibility of restarting the model with the assimilated field.

**This paper makes the following contributions:**

1. We propose a new data assimilation algorithm that works with multiple geophysical fields by leveraging VAE architecture with self-attention layers in the latent space.

2. We test our algorithm on different setups and show that it outperforms the baselines: classical 3D-VAR and a VAE-based approach (Melinc & Zaplotnik, 2024).

3. We integrated our algorithm inside the operational forecasting ocean model Nucleus for European Modeling of the Ocean (NEMO) to assimilate real sattelite observations. We have shown that neural network based data assimilation improves the forecasts quality.

## 2 DATA

The area of interest for this work is the Barents Sea and the Kara Sea (see Figure 1). The Kara and Barents Seas are not covered with multi-year ice de Gelis et al. (2021). For the selected region of the Kara Sea, the freeze-up begins rapidly in December, with melting in April. The selected region of the Barents Sea has a completely open sea throughout most of the year.

## 3 MODEL

### 3.1 BACKGROUND

As an assimilation background we use ocean data that are generated by our production numerical forecasting system that consists of numerical Weather Research and Forecasting model (WRF) and Nucleus for European Modeling of the Ocean (NEMO) of version 4.0 with integrated Sea Ice Modeling Integrated Initiative (SI3). Spatial resolution of the modeling dataset is around 3-4 km, we

used ORCA12 grid and Drakkar configuration that is considered state of the art for high resolution ocean modeling. Since NEMO requires atmosphere forcing we also used these data as additional features (these data are generated by the atmosphere model Weather Research and Forecasting WRF of version 4.4). For the period of interest, the dataset contains daily prediction of the next 72 hours for ocean and atmosphere variables. Data range from 2015 to 2023. In this study, we focus on three key fields: the sea ice concentration field (where data assimilation is performed), the sea ice thickness field, and the sea ice temperature field. Data from 2015 to 2021 (inclusive) are used to compute statistical properties and train the variational autoencoder (VAE). 2022 and 2023 are reserved for validating the accuracy of field reconstruction and conducting data assimilation experiments.

## 3.2 Observations

The following observational data was considered:

1. Sentinel-3 Altimetry (SRAL) Sea Ice Thematic Product Aublanc et al. (2025). The dataset includes satellite tracks with an along-track resolution of up to 330 meters. The variable "sea_ice_concentration_20_ku" provides ice concentration data and is available as an auxiliary product. The primary data source is the OSI-430-a product, which has a 2-day latency and a spatial resolution of 25 km. Therefore, greater scientific interest lies in the variable "surf_type_class_20_ku", which characterizes surface type (open ocean, floes, lead, unclassified) based on sea ice concentration and waveform peakiness, and the variable "freeboard_20_ku" (radar freeboard). In future product versions, the latter is planned to be converted into ice thickness.

2. ASI AMSR2 Spreen et al. (2008) sea ice concentration data. These data have a spatial resolution of 6.25 km and are available in near-real time (NRT) with a latency about 25 hours. A key advantage of AMSR2 is its all-weather, day-and-night operational capability, enabled by passive microwave measurements. However, a known limitation is the reduced accuracy in sea ice concentration retrieval during melt seasons due to changes in surface emissivity caused by snow and ice melt Pang et al. (2018).

We have determined that the most promising problem to address is the assimilation of data tracks, especially since Sentinel-3 is expected to enhance its thematic product on sea ice with ice thickness information. For model-to-model assimilation, we use sea ice concentration data from a different model year, selected along SRAL tracks. For satellite-to-model assimilation, we employ AMSR2 sea ice concentration data along SRAL tracks, adjusted using the surf_type_class_20_ku flag: values are set to zero for the "open ocean" class, while for "floes" and "lead" classes with zero concentration, the nearest non-zero value is assigned.

## 3.3 Validation

For model-to-model assimilation, the model fields corresponding to the dates of the assimilation tracks are used for validation and comparison. For satellite-to-model assimilation, AMSR2 data serve as the validation data set.

## 3.4 Data analysis

To compare sea ice concentration data from different sources along the Sentinel-3 track between December 2019 and April 2020, their cumulative distribution functions (CDF) should be analyzed (Fig. 2). Data were collected for identical dates and identical spatial domains in all sources. The NEMO model (blue line) exhibits a lower probability of zero ice concentration compared to AMSR2 (green line), indicating systematic overestimation of ice cover in the model output. The SRAL-derived 'sea_ice_concentration_20_ku' variable (orange line) shows markedly different slope characteristics than both NEMO and AMSR2. This divergence stems from the coarse resolution of OSI-430-a product (source of this variable), which introduces spatial averaging effects that increase the number of intermediate values.

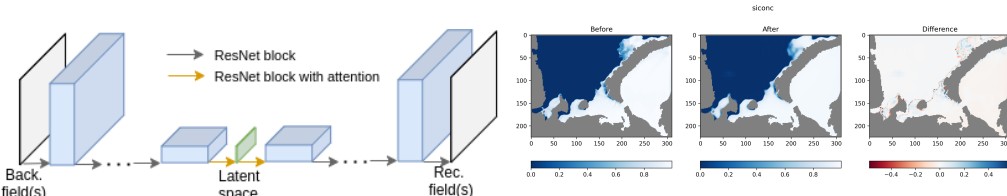

Figure 3: Variational Autoencoder architecture.     Figure 4: Example of VAE data reconstruction.

## 4 MODEL

In the 3D-VAR algorithm, spatial relationships between the components of a field, as well as inter-field correlations, are represented in the background error covariance matrix. In this study, we propose using a variational autoencoder (Kingma & Welling, 2014) to capture and represent these patterns. A related approach was previously applied to temperature data at the 850-hPa pressure level from the ERA5 reanalysis, as reported in (Melinc & Zaplotnik, 2024). We used the VAE architecture from (Melinc & Zaplotnik, 2024) as our baseline, denoting it as $base\_vae\_1f$. In our work, we employ a different autoencoder architecture and perform experiments using both single-field and multi-field datasets. The architecture we propose is inspired by the VAE architectures widely used in stable diffusion models Rombach et al. (2022), which have demonstrated strong generative and representation learning capabilities. The main characteristics of our architecture 3 are as follows: (1) the latent space is implemented as a feature map rather than a vector, (2) ResNet-based blocks are used throughout the encoder and decoder, and (3) attention mechanisms are introduced in the middle layers to enhance feature interaction.

We conducted experiments with a date-conditioned autoencoder where: dates were converted to day-of-year (DOY) values. DOY was encoded as a cyclic variable using sine/cosine transformation: $(\sin(2\pi \cdot DOY/366), \cos(2\pi \cdot DOY/366))$ which are transformed by the linear layer and concatenated with in the latent space.

### 4.1 TRAINING

The variational autoencoder learns to reconstruct fields using the reparameterization trick. Mean squared error (MSE) is used as the reconstruction loss, while the KL divergence is computed for the Gaussian case (Kingma & Welling, 2014). We also experimented with the addition of a discriminator and SSIM loss, but it did not produce significant improvements.

We used the Lion optimizer (Chen et al., 2023) for training, as it demonstrated a more stable convergence in our experiments. Missing data (over land) are filled with physically adequate values.

### 4.2 ASSIMILATION

For the baseline data assimilation experiment, we employed a 3D-VAR scheme $3d\_var$. The background error covariance matrix B was modeled using a fifth-order quasi-Gaussian function (Gaspari & Cohn, 1999) with a length scale of 100 km.

The latent space assimilation algorithm (Algorithm 1) is presented below. The key feature of the algorithm: assimilation is performed in the autoencoder's latent space. The latent space field is adjusted via backpropagation to ensure the reconstructed (analysis) field better approximates the observational data.

It is important to discuss the features of the $Loss$ function. The loss function consists of three terms: (1) the observation error, (2) the background field error, and (3) the latent-space error. The second and third terms serve as regularizations to prevent a significant deviation from the model's original field and are assigned smaller weighting coefficients.

$$Loss(x_a, y, x_b, z, z_0) = w_y MSE(H(x_a), y)$$
$$+ w_b MSE(x_a, x_b) + w_z MSE(z, z_0) \tag{3}$$

---

**Algorithm 1** Latent Space Assimilation (LSA)

---

**Require:** $Encoder$ and $Decoder$ - part of pretrained VAE, $Loss$ - error function
    **Input:** $x_b$ - background fields, $y$ - observations
    **Output:** $x_a$ - analysis fields

    Encode background: $z_0 \leftarrow Encoder(x_b)$
    Initialize optimizable latent: $z \leftarrow z_0$ (requires gradient)

    **for** iteration i=0 to N **do**
        Decode current latent: $x_a \leftarrow Decoder(z)$
        Compute loss: $L_{total} \leftarrow Loss(x_a, y, x_b, z, z_0)$
        Compute gradient: $\nabla_z L_{total}$
        Update latent: optimization step
    **end for** **return** $x_a \leftarrow$ Decoder($z$)

---

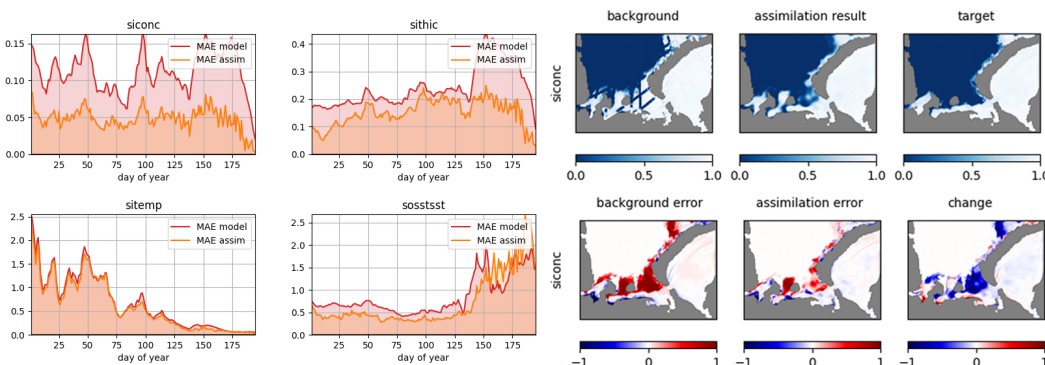

Figure 5: MAE metric values for $vae\_4f$ model from different physical fields by day of the year on the left. Sea Ice Concentration assimilation example for 12.04 using $vae\_4f$ model on the right.

Given the slow evolution of ice concentration (typically $< 1\%$ daily change), we tested assimilation of the prior three days data.

## 5 EXPERIMENTS

Since we aimed at improving the quality of our forecasting system in the real world through sentinel 3 SRAL assimilation, we organized our framework as follows. First, we evaluated the quality of the VAE reconstruction. Secondly, we experimented with model-to-model assimilation to tune out probable overfitting and deal with general architecture search. After that we tested the best resulting models on real-world data and tested our assimilation model inside NEMO forecasting pipeline.

The VAE should reconstruct the fields reasonably well, though perfect accuracy is not the goal, as VAEs inherently balance reconstruction fidelity against the dimensionality of the compressed representation (i.e., latent space sparsity). The goal of the model-to-model assimilation is to estimate the quality of assimilation in neutral setting. Since observation data and model data are noisy and usually distributed differently, thus introducing bias in the quality of assimilation. Model-to-model, on the other hand, demonstrates the quality of assimilation in the physics-driven process that is represented by the numerical model inside NEMO. For the forecasting pipeline, let's call this assimilation scheme production data assimilation.

### 5.1 RECONSTRUCTION ERROR

The model was trained on data from 2015 to 2021 and 2022 was used for validation. The following fields were utilized: siconc (sea ice concentration), sithic (sea ice thickness), sitemp (sea

ice temperature) and sosstsst (sea surface temperature). An example of sea ice concentration field reconstruction is shown in Figure 4. The results are presented in Table 1 .

The evaluation metrics used are MAE (Mean Absolute Error) and MSE (Mean Squared Error). MAE estimates the average error magnitude, while MSE penalizes larger errors more heavily and smaller errors less severely due to its quadratic nature. The results are presented in Table 1. Features of the architecture and naming of models: _*f - shows how many physical fields are fed to the model input. _d* the latent space is transformed by a linear layer from a feature map into a vector with the specified number of features. _m the sitemp field is replaced by the sosstsst field, _emb - a condition for the day of the year is added, _c* - the number of feature maps in the latent space, 1 by default.

Table 1: VAE reconstruction quality assessment

| Model name | siconc MAE | sithic MAE | sitemp MAE | sosstsst MAE |
|---|---|---|---|---|
| base_vae_1f | $0.028 \pm 0.002$ | - | - | - |
| vae_1f | $0.008 \pm 0.001$ | - | - | - |
| vae_1f_d512 | $0.023 \pm 0.002$ | - | - | - |
| vae_3f | $0.023 \pm 0.002$ | $0.083 \pm 0.006$ | $0.094 \pm 0.007$ | - |
| vae_3f_emb | $0.026 \pm 0.002$ | $0.090 \pm 0.007$ | $0.093 \pm 0.008$ | - |
| vae_3f_m | $0.015 \pm 0.001$ | $0.024 \pm 0.002$ | - | $0.252 \pm 0.006$ |
| vae_3f_m_emb | $0.016 \pm 0.001$ | $0.026 \pm 0.002$ | - | $0.254 \pm 0.007$ |
| vae_4f | $0.024 \pm 0.002$ | $0.083 \pm 0.006$ | $0.085 \pm 0.007$ | $0.265 \pm 0.008$ |
| vae_4f_emb | $0.032 \pm 0.002$ | $0.099 \pm 0.008$ | $0.083 \pm 0.007$ | $0.267 \pm 0.008$ |
| vae_4f_c2 | $0.018 \pm 0.001$ | $0.029 \pm 0.002$ | $0.062 \pm 0.005$ | $0.210 \pm 0.005$ |
| vae_4f_c2_emb | $0.017 \pm 0.001$ | $0.031 \pm 0.002$ | $0.061 \pm 0.005$ | $0.186 \pm 0.003$ |

## 5.2 QUALITY ESTIMATION

### 5.2.1 MODEL TO MODEL ASSIMILATION

In this experiment, our objective was to analyze the ability of a trained VAE to capture relationships between values and physical fields. Unfortunately, we do not know the true field values at each time step: The model contains computational errors and depends on the initial initialization, which also introduces uncertainties. Satellite data have measurement errors and interpretation challenges.

Therefore, we conclude that it is necessary to verify the assimilation of the model data into the model data. We sample see ice concentration data along the SRAL tracks from the NEMO forecast for next year $\tilde{y} = x_{i+365}[mask]$ and assimilate them into current data $x_i$. To evaluate performance, we compare the post-assimilation fields with the prediction of the model $x_{i+365}$ (see Algorithm 2). Since we treat NEMO numerical modeling as physics-based, we expect that a higher quality model will assimilate data closer to the NEMO output.

---

**Algorithm 2** Model-to-model (M2M) data assimilation

---

**Require:** $\{x_1, x_2, ..., x_n\}$ - NEMO simulation fields, $\{y_1, y_2, ..., y_n\}$ - AMSR corrected by SRAL data
    **for** day in assimilation cycle **do**
        Create mask: $mask = y_i$ is not None
        Sample observation: $\tilde{y}_i = x_{i+365}[mask]$
        Make assimilation: $x_i^a = \text{LSA}(x_i, \tilde{y}_i)$
        Make validation: $x_i$ and $x_i^a$ vs $x_{i+365}$
    **end for**

---

Examples of model-in-model assimilation for $3d\_var\ vae\_4f$ are illustrated in figures in 7. Only sea ice concentration values are assimilated. It can be seen that $vae\_4f$ not only yields a lower error but also produces a sharper ice-water boundary, while $3d\_var$ tends to smooth it. The $vae\_4f$ results show that when the concentration of ice decreases, the thickness of the ice in the same region also

Table 2: Model to model assimilation result: average MAE

| Model name | siconc | sithic | sitemp | sosstsst |
|---|---|---|---|---|
| background | $0.112 \pm 0.002$ | $0.242 \pm 0.005$ | $0.67 \pm 0.04$ | $0.86 \pm 0.03$ |
| 3d_var | $0.052 \pm 0.001$ | - | - | - |
| base_vae_1f | $0.080 \pm 0.002$ | - | - | - |
| vae_1f | $0.051 \pm 0.001$ | - | - | - |
| vae_1f_d512 | $0.053 \pm 0.001$ | - | - | - |
| vae_3f | $0.060 \pm 0.001$ | $0.172 \pm 0.004$ | $0.63 \pm 0.04$ | - |
| vae_3f_emb | $0.057 \pm 0.001$ | $0.167 \pm 0.004$ | $0.64 \pm 0.04$ | - |
| vae_3f_m | $0.061 \pm 0.001$ | $0.200 \pm 0.004$ | - | $0.84 \pm 0.05$ |
| vae_3f_m_emb | $0.062 \pm 0.001$ | $0.213 \pm 0.004$ | - | $0.811 \pm 0.045$ |
| vae_4f | $\mathbf{0.0481 \pm 0.0009}$ | $\mathbf{0.152 \pm 0.003}$ | $0.615 \pm 0.036$ | $0.732 \pm 0.039$ |
| vae_4f_emb | $0.0493 \pm 0.0009$ | $0.158 \pm 0.004$ | $\mathbf{0.608 \pm 0.035}$ | $\mathbf{0.663 \pm 0.031}$ |
| vae_4f_c2 | $0.0508 \pm 0.0010$ | $0.168 \pm 0.003$ | $0.616 \pm 0.035$ | $0.739 \pm 0.034$ |
| vae_4f_c2_emb | $0.0504 \pm 0.0009$ | $0.176 \pm 0.003$ | $\mathbf{0.608 \pm 0.035}$ | $0.724 \pm 0.037$ |

decreases, while both the temperature of the ice and the temperature of the ocean increase. These patterns align with the target fields and maintain physical consistency with the model. This clearly shows that the VAE successfully captures not only relationships within individual physical variables but also the cross-correlations between different physical parameters.

Figure 5 shows the MSE and MAE metric values for data assimilation using the $vae\_4f$ model, applied to the fields of ice concentration, ice thickness, ice temperature and upper-layer water temperature over different days of the year. The data is interrupted in mid-July when the ice in the Barents and Kara Seas completely melts. It is evident that the ice concentration and thickness fields are well-correlated, whereas the ice and water temperature fields show weaker dependence on the ice concentration. Furthermore, Table 2 present the MAE metrics for assimilation across different models. We observe that the incorporation of temperature fields improves the accuracy of the ice concentration and thickness assimilation. The $vae\_4f$ model was selected as the primary because of its best performance in capturing the relationship between ice concentration and thickness.

### 5.2.2 SAT TO MODEL ASSIMILATION

The next step was data assimilation of AMSR2 ice concentration data corrected by SRAL surface type. We evaluated the quality of assimilation against the AMSR2 data and corrected AMSR2 data that had not yet been assimilated. The assimilation scheme works as follows.

---

**Algorithm 3** Satellite-to-model (S2M) data assimilation

---

**Require:** $\{x_1, x_2, ..., x_n\}$ - NEMO simulation fields, $\{y_1, y_2, ..., y_n\}$ - AMSR corrected by SRAL data, $\{v_1, v_2, ..., v_n\}$ - AMSR data.
    **for** day in assimilation cycle **do**
        Make assimilation: $x_i^a = \text{LSA}(x_i, y_i)$
        Make validation: $x_i$ and $x_i^a$ vs $v_i$ and $y_{i+1}$
    **end for**

---

The assimilation results for the SRAL-corrected AMSR2 sea ice concentration data are presented in the Table 3. Although the $vae\_3f\_emb$ model showed slightly better metrics, we selected the $vae\_4f$ model for the final experiment. This decision is based on its superior performance in capturing ice thickness variations in the M2M experiment, coupled with the fact that the difference in results for this experiment is within the margin of error. Samples of assimilation results are in Appendix A.3.

### 5.3 PRACTICAL APPLICATION

In the last experiment, we integrate the results of the satellite data assimilation into the NEMO pipeline using the restart mechanism. The model restart-ice fields were modified using the $sicon$ (sea ice concentration) and $sithic$ (sea ice thickness) fields, which assimilated satellite data via the

Table 3: Satellite to model assimilation result

| Model | AMSR2 | | AMSR2 corrected (track) | |
|---|---|---|---|---|
| name | MSE | MAE | MSE | MAE |
| background | $0.027 \pm 0.002$ | $0.049 \pm 0.002$ | $0.028 \pm 0.002$ | $0.050 \pm 0.003$ |
| 3d_var | $0.013 \pm 0.001$ | $0.037 \pm 0.002$ | $0.019 \pm 0.002$ | $0.044 \pm 0.002$ |
| base_vae_1f | $0.019 \pm 0.001$ | $0.045 \pm 0.002$ | $0.021 \pm 0.002$ | $0.051 \pm 0.003$ |
| vae_1f | $0.014 \pm 0.001$ | $0.034 \pm 0.002$ | $0.019 \pm 0.002$ | $0.040 \pm 0.002$ |
| vae_1f_d512 | $\mathbf{0.012 \pm 0.001}$ | $0.033 \pm 0.002$ | $\mathbf{0.017 \pm 0.002}$ | $0.041 \pm 0.002$ |
| vae_3f | $\mathbf{0.012 \pm 0.001}$ | $0.033 \pm 0.002$ | $\mathbf{0.017 \pm 0.002}$ | $0.040 \pm 0.002$ |
| vae_3f_emb | $\mathbf{0.012 \pm 0.001}$ | $\mathbf{0.032 \pm 0.002}$ | $\mathbf{0.017 \pm 0.002}$ | $\mathbf{0.039 \pm 0.002}$ |
| vae_3f_m | $0.014 \pm 0.001$ | $0.034 \pm 0.002$ | $0.018 \pm 0.002$ | $0.043 \pm 0.002$ |
| vae_3f_m_emb | $0.013 \pm 0.001$ | $0.036 \pm 0.002$ | $0.018 \pm 0.002$ | $0.044 \pm 0.002$ |
| vae_4f | $0.013 \pm 0.001$ | $0.033 \pm 0.002$ | $0.018 \pm 0.002$ | $0.041 \pm 0.002$ |
| vae_4f_emb | $\mathbf{0.012 \pm 0.001}$ | $0.039 \pm 0.002$ | $0.018 \pm 0.002$ | $0.046 \pm 0.002$ |
| vae_4f_c2 | $0.014 \pm 0.001$ | $0.033 \pm 0.002$ | $0.019 \pm 0.002$ | $0.040 \pm 0.002$ |
| vae_4f_c2_emb | $0.013 \pm 0.001$ | $0.033 \pm 0.002$ | $0.019 \pm 0.002$ | $0.041 \pm 0.002$ |

$vae\_4f$ model. The $sitemp$ (ice surface temperature) and $sosstsst$ (sea surface temperature) fields were deliberately excluded, as they demonstrated a weaker correlation with the assimilated values. The restart data is modified as told in appendix A.1.

The experiment was organized as follows. A specific date was selected. For this date, data assimilation was performed and the restart file was modified. Subsequently, a 5-day forecast was run using the NEMO model. The forecast results were then compared with the corresponding AMSR2 data for those days. The outcome of this comparison is presented in the Figure 8. Metrics for the experiments are in Table 4.

The experiment demonstrated the feasibility of using fields processed by neural network-based data assimilation within the NEMO computational model for forecasting. The predictions generated by this approach are physically consistent and show good agreement with satellite observations.

Table 4: Application result: MAE with AMSR2 (20-02-2025)

| Experiment | 1 day | 2 day | 3 day | 4 day | 5 day |
|---|---|---|---|---|---|
| model | 0.142 | 0.123 | 0.081 | 0.084 | 0.081 |
| model+assimilation | 0.079 | 0.086 | 0.063 | 0.072 | 0.072 |

## 6 CONCLUSION

This work is devoted to multi-field data assimilation by neural networks as an alternative to classical approaches. For this we used a variational autoencoder with self-attention layers that helped us to account to cross-corelation between the physics fields. We have shown that such an approach surpasses in terms of quality both classical approach 3D-VAR and the competing approaches from literature based on similar ideas.

Also we tested our approach on the real operative sea ice forecasting workflow in Arctic based on Nucleus for European Modelling of the Ocean (NEMO) state-of-the-art oceam modeling framework. We have shown that assimilation of real sattelite data in the model quickly improves the model quality, so our approach has strong practical applications.

Possible direction of our approach improvement would be to correct not only a single snapshot but a whole time series in order to assure physical consistency of the forecasts dynamics. The may challenge here lies in learning a good evolution operator for the whole bunch of participating physical fields. In order to improve quality of forecasts even further it would make sense not only to assimilate sea ice data but also to correct atmosphere forcing and increase domain of simulations to capture more oceanic and atmospheric processes in the Arctic.

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

## A  APPENDIX

### A.1  ALGORITHM OF MODIFICATION OF RESTARTS IN THE NEMO OCEAN MODEL

With initial restart files, written by the ocean model and new assimilated fields that came out of assimilation algorithm following was applied:

1. The *sicon* concentration values were clipped to the [0, 1] range to ensure physical consistency. Values below 0.01 were treated as zero, and a mask was created based on this threshold to distinguish ice from open water. The corrected *sicon* data is written to the $a\_i$ variable.

2. Ice thickness and concentration are used to calculate the ice volume per unit area $v\_i$, using the formula $v\_i = a\_i \cdot sithic$.

3. Since snow cannot lie on water, the snow volume per unit area ($v\_s$) and the ice concentration ($a\_i$) from the restart file are used to calculate the snow thickness. This snow thickness is then multiplied by the new ice concentration ($a\_i$): $v\_s = \frac{v\_s[rest] \cdot a\_i}{a\_i[rest]}$.

4. The *snwice_mass* and *snwice_mass_b* values were recalculated from $v\_i$ and $v\_s$ using the average densities of sea ice and snow.

5. The salinity variable $sv\_i$ is recalculated for the new ice volume $v\_i$. A value of 7.4 psu was assigned for newly formed ice.

6. The values of $e\_s\_l01$, $e\_i\_l01$, and $e\_i\_l02$ are recalculated proportionally to $v\_s$ and $v\_i$ taking into account the heat capacity and latent heat of fusion for snow and ice.

7. The internal stress variables $stress1\_i$, $stress2\_i$, and $stress12\_i$ are recalculated proportionally to the ice volume $v\_i$.

8. The $oa\_i$ value is scaled proportionally with $a\_i$.

9. The ocean state components of the restart file were left unmodified, as the change in ice concentration was not substantial.

## A.2   Additional assimilation process illustrations

These are examples of assimilated sea ice fields for model-to-model assimilation experiments and sat-to-model experiments.

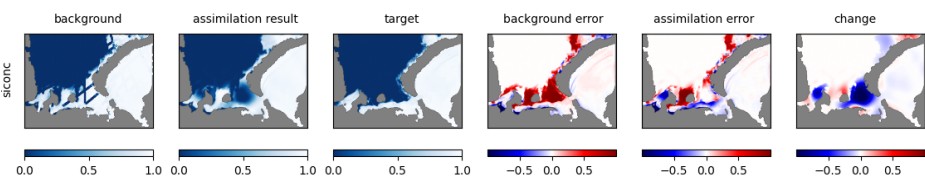

Figure 6: Results M2M assimilation for 12 April with use $3d\_var$ model. sicinc − sea ice concentration. Columns: background − initial values, assimilation result − M2M output, target − reference values, background error = background − target (initial field deviation), assimilation error = assimilation result − target (post assimilation deviation), change = assimilation result − background (assimilation induced adjustment).

## A.3   The Use of Large Language Models (LLMs)

The text in the paper was initially written by humans and then sent to LLM in the process to suggest stylistic improvements and to correct grammar and punctuation mistakes.

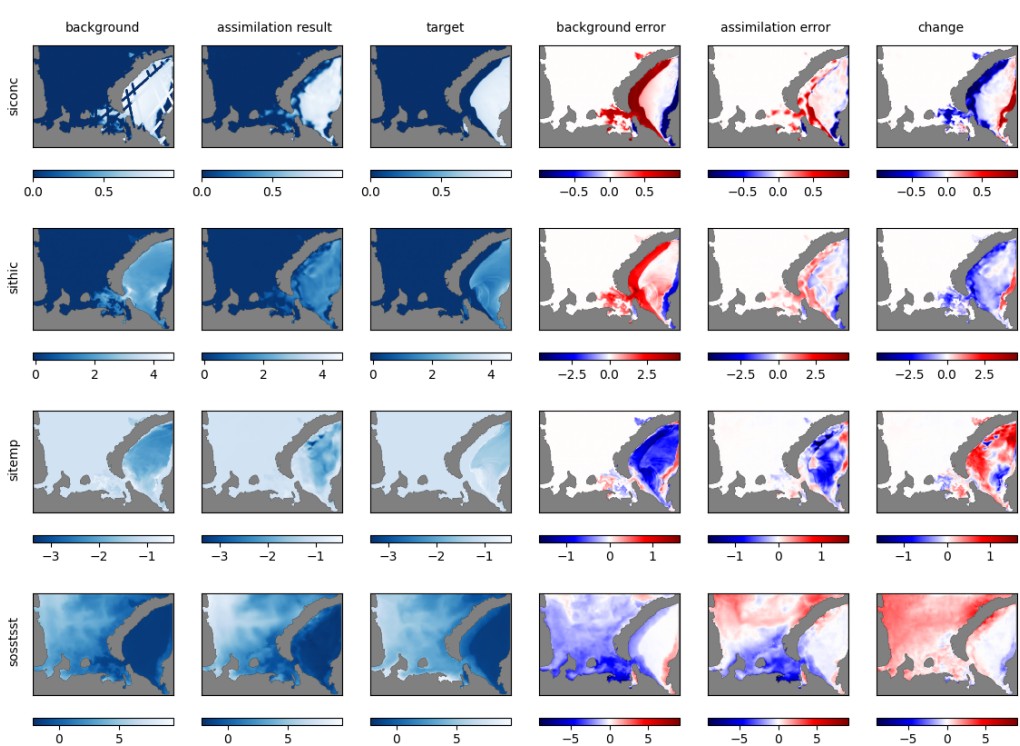

Figure 7: Results M2M assimilation for 30 May with use $vae\_4f$ model. Rows: siconc − sea ice concentration, sithic − sea ice thickness, sosstsst − sea surface temperature. Columns: background − initial values, assimilation result − M2M output, target − reference values, background error = background − target (initial field deviation), assimilation error = assimilation result − target (post assimilation deviation), change = assimilation result − background (assimilation induced adjustment).

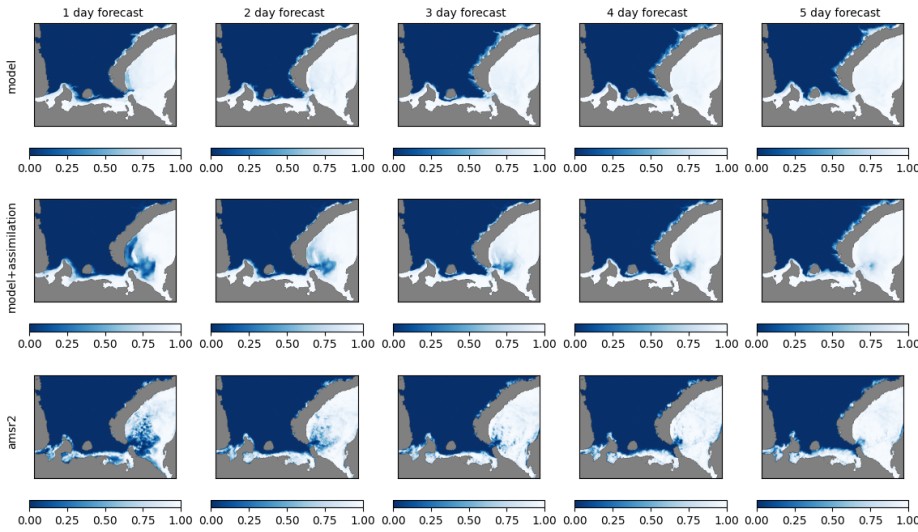

Figure 8: NEMO model prediction results. The forecast was initialized on February 22, 2023, and run for 5 days. Top row: Ice concentration values predicted by the model without data assimilation. Middle row: Ice concentration values predicted by the model with data assimilation (assimilation was performed only on the initial day). Bottom row: Reference AMSR2 satellite ice concentration data for comparison.

