# OpenReview forum: "Multi-field sea ice data assimilation with a variational autoencoder"
_ICLR.cc/2026/Conference — ICLR 2026 Conference Withdrawn Submission_

### Official Review · Reviewer_DJHv · 2025-10-23

**Soundness:** 1
**Presentation:** 3
**Contribution:** 1
**Rating:** 2
**Confidence:** 4

**Summary:**

The paper tackles sea-ice data assimilation and proposes a VAE-based latent assimilation framework (with self-attention) to implicitly model covariance and perform multivariate DA. A fairly extensive set of experiments on a sea-ice dataset suggests the approach is effective. However, the methodological novelty feels incremental, largely combining known components and extending prior univariate work, while positioning against recent diffusion- and VAE-based DA and attention-centric weather models is incomplete. Important implementation details remain unclear (e.g., the exact sea-ice-tailored network design and how the 3DVar background covariance $B$ is built), and the evaluation lacks strong, head-to-head baselines. Overall, this reads as a solid domain study for sea-ice DA; the contribution would be strengthened by clearer architectural justification, fuller literature coverage, and competitive comparisons.

**Strengths:**

This paper addresses sea-ice data assimilation, an important and challenging problem in meteorology. By employing a VAE for latent-space assimilation, it implicitly represents the covariance structure, marking a meaningful step toward AI-driven data assimilation in this domain. The authors also present a fairly extensive suite of assimilation experiments that provide evidence for the method’s effectiveness.

**Weaknesses:**

- **Novelty & positioning are limited.** The method largely combines known ideas, including latent data assimilation and a VAE augmented with self-attention, and extends [1] from univariate to multivariate. Given recent multivariate and even real-model results in AI data assimilation (e.g., diffusion-based [2] and VAE-based [3]), plus the fact that attention for cross-variable structure is now standard (e.g., Pangu-Weather [4]), the contribution feels incremental. The paper also lacks strong head-to-head baselines against these lines, making its advantages unclear.
- **Scope suggests a domain study rather than an ICLR-level advance.** The primary value appears to be a careful analysis on a sea-ice dataset, which is interesting for the meteorology community but reads more like a domain-specific study than a broadly novel algorithmic contribution appropriate for ICLR.

[1] Melinc, Boštjan, and Žiga Zaplotnik. "3D‐Var data assimilation using a variational autoencoder." *Quarterly Journal of the Royal Meteorological Society* 150.761 (2024): 2273-2295.

[2] Huang, Langwen, et al. "Diffda: a diffusion model for weather-scale data assimilation." *arXiv preprint arXiv:2401.05932* (2024).

[3] Xiao, Yi, et al. "VAE-Var: Variational autoencoder-enhanced variational methods for data assimilation in meteorology." *The Thirteenth International Conference on Learning Representations*. 2025.

[4] Bi, Kaifeng, et al. "Accurate medium-range global weather forecasting with 3D neural networks." *Nature* 619.7970 (2023): 533-538.

**Questions:**

- Could you describe the neural network architecture in detail, for example, how it’s tailored to sea-ice data (e.g., inputs/targets, spatiotemporal inductive biases, physical constraints) and why it outperforms alternative designs such as vision transformers?
- Coud you elaborate on the baseline implementations, especially 3DVar: how is the background-error covariance $B$ specified or estimated (e.g., climatology, ensemble-derived, localization/balancing)?
- Could you clarify how the weights $w_y, w_b, w_z$ in Equation (3) were selected and provide any sensitivity/ablation evidence, given their likely impact on assimilation performance?

---

### Official Review · Reviewer_Ye3A · 2025-10-28

**Soundness:** 1
**Presentation:** 2
**Contribution:** 1
**Rating:** 0
**Confidence:** 3

**Summary:**

This study proposes a neural data assimilation system using a variational autoencoder (VAE) to improve sea ice forecasts in high-resolution ocean models. Multiple physical fields are combined with the incorporation of pixel-wise self-attention, and the model captures complex spatial and cross-field relationships. The author validates the approach with real satellite and ocean model data. Their approach effectively handles sparse, noisy observations and reduces sea ice concentration errors. They also tested their approach by integrating it into a real-time NEMO ocean model operational forecasting system. The work provides a scalable, non-Gaussian alternative to traditional 3D-VAR assimilation methods.

**Strengths:**

S1. The proposed approach uses a multi-field assimilation technique that simultaneously processes several important sea ice and ocean fields.

S2. I consider this article to have strong potential that needs to be improved for clarity and novelty, because the integration of its algorithm inside the operational forecasting ocean model Nucleus for European Modelling of the Ocean (NEMO) to assimilate real satellite observations presents a real-world use-case scenario.

S3. The chosen Pan-Arctic basins play an important role in understanding sea ice thickness and sea ice concentration. Their choice of region is a good fit.

**Weaknesses:**

W1: They discuss several model data and satellite data usages. However, they do not explain well how their proposed data assimilation approach is comparable to the classical one. Also, there is no novelty in the cost function, as it is directly derived from the 3D-Var approach.

W2:  Missing novelty in the architecture and specific reason for using ResNet in the decoder.
The baselines are not chosen carefully (there have been several studies relevant to data assimilation and neural networks).

W3: The improvement in performance cannot be explained simply by adding features, because the base method of the proposed work did not show much difference compared to the chosen baselines.

W4: I am skeptical about the performance and evaluation metrics, as they do not represent improved performance.
Moreover, the presented algorithm signifies nothing; it does not convey any meaningful improvement.

W5: It is questionable that they claim the 4F model predicts better; however, the performance metrics of the 4F model are almost the same as those of the classical 3D-Var method.

W6: In Fig. 5, the sea ice concentration and sea ice thickness errors of 0.1 and 0.5 are still significant when compared with the performance metrics of classical methods. The shift in assimilation does not necessarily indicate a reduction in error; rather, it suggests a large error introduced by the assimilation process.

**Questions:**

Q1: If the function is not novel, the authors must explicitly state why the fusion of this classical cost function with the proposed architecture is a novel design choice.

Q2: The paper must include comparisons against more recent and relevant methods, particularly those that address data sparsity using neural networks (e.g., other physics-constrained neural networks, or advanced filtering techniques like Particle Filters or Hybrid EnKF/3D-Var approaches) to clearly delineate the contribution.

Q3: According to Figure 5, there is no significant pattern association between temperature fields and sea ice fields. This contradicts the prediction perspectives, which are important to be preserved during the assimilation step.

---

### Official Review · Reviewer_Mksy · 2025-10-30

**Soundness:** 2
**Presentation:** 2
**Contribution:** 2
**Rating:** 2
**Confidence:** 3

**Summary:**

This paper presents a novel neural data assimilation system based on an enhanced Variational Autoencoder (VAE) for improving high-resolution sea ice forecasts in the NEMO ocean model.

**Strengths:**

1.Effectively assimilates sparse and noisy real-world satellite observations (Sentinel-3 SRAL, AMSR2), significantly reducing forecast errors.

2.Offers a scalable, non-Gaussian alternative to traditional data assimilation methods (like 3D-VAR), which is crucial for fields with non-Gaussian error distributions (e.g., sea ice concentration).

**Weaknesses:**

1.While the paper uses a modern VAE (Variational Autoencoder) with enhancements like ResNet blocks and pixel-wise self-attention, the core idea of using a VAE for data assimilation (specifically the concept of an observation operator and prior modeling) is a known technique in the field of deep learning for geosciences (e.g., in previous work like VAE-DA). The innovation lies primarily in the specific engineering and application of this architecture to a multi-field sea ice problem, rather than a fundamentally new theoretical data assimilation framework.

2.The primary comparison is against 3D-VAR (a static, batch-based method). Comparing the performance and computational efficiency against more advanced or state-of-the-art sequential assimilation methods (like Ensemble Kalman Filters (EnKF) or Particle Filters), especially those designed for non-Gaussian/non-linear systems, would better highlight the unique innovative advantage of the VAE approach in the broader data assimilation landscape.

3.Experiments are limited to the Barents and Kara Sea region and are only relevant during the ice season, meaning further validation is needed for other sea ice regimes and characteristics.

**Questions:**

1.What physical insight can be derived from the VAE's latent space? Does the latent representation clearly correspond to major modes of sea ice variability, and can it be used for meaningful physical diagnostics or error characterization?

2.The paper highlights the use of pixel-wise self-attention. Can the authors provide a more in-depth interpretation or visualization of the attention weights to demonstrate how the network leverages spatial and cross-field correlations during the assimilation step?

3.The VAE acts as a non-linear, non-Gaussian prior. Can the authors quantify the separate contribution of this learned prior versus the improved observation operator to the overall assimilation skill?

---

### Author Response · Authors · 2025-11-26
**Thanks for insightful comments! We decided to withdraw the paper**

Dear reviewers,

We thank you for your valuable input and will try to improve the article considering your comments when preparing it for some another venue. As for now we understand that chances for it to pass are negligibly small and we decided to withdraw the paper in order to save your time.

Best wishes,
the authors

---

### Note · Authors · 2025-11-26

I have read and agree with the venue's withdrawal policy on behalf of myself and my co-authors.